# Spatial Distribution of Muscular Effects of Acute Whole-Body Electromyostimulation at the Mid-Thigh and Lower Leg—A Pilot Study Applying Magnetic Resonance Imaging

**DOI:** 10.3390/s222410017

**Published:** 2022-12-19

**Authors:** Marina Götz, Rafael Heiss, Simon von Stengel, Frank Roemer, Joshua Berger, Armin Nagel, Michael Uder, Wolfgang Kemmler

**Affiliations:** 1Institute of Medical Physics, Friedrich-Alexander-University of Erlangen-Nürnberg, 91052 Erlangen, Germany; 2Institute of Radiology, University Hospital Erlangen, 91054 Erlangen, Germany; 3German University for Prevention and Health Management, 66123 Saarbrücken, Germany

**Keywords:** electrical muscle stimulation, neuromuscular electrical myostimulation, edema, MRI, spatial distribution, lower extremities

## Abstract

Whole-body electromyostimulation (WB-EMS) is an innovative training method that stimulates large areas simultaneously. In order to determine the spatial distribution of WB-EMS with respect to volume involvement and stimulation depth, we determined the extent of intramuscular edema using magnetic resonance imaging (MRI) as a marker of structural effects. Intense WB-EMS first application (20 min, bipolar, 85 Hz, 350 µs) was conducted with eight physically less trained students without previous WB-EMS experience. Transversal T2-weighted MRI was performed at baseline and 72 h post WB-EMS to identify edema at the mid-thigh and lower leg. The depth of the edema ranged from superficial to maximum depth with superficial and deeper muscle groups of the mid-thigh or lower leg area approximately affected in a similar fashion. However, the grade of edema differed between the muscle groups, which suggests that the intensity of EMS-induced muscular contraction was not identical for all muscles. WB-EMS of the muscles via surface cuff electrodes has an effect on deeper parts of the stimulated anatomy. Reviewing the spatial and volume distribution, we observed a heterogeneous pattern of edema. We attribute this finding predominately to different stimulus thresholds of the muscles and differences in the stress resistance of the muscles.

## 1. Introduction

Whole-body electromyostimulation (WB-EMS) is a recognized time-efficient and joint-friendly training technology with increasing popularity [1]. Most importantly, WB-EMS differs from the related local neuromuscular (N)EMS technology [2] in the application of at least six current channels and corresponding participation of all major muscle groups (with dedicated impulse intensity) [3]. Considering present WB-EMS application as a resistance-type exercise [4], predominantly musculoskeletal parameters related to muscle strength, mass or low back pain were addressed. However, negative aspects of a technology able to stimulate large muscle groups simultaneously and, in excess, with supra-maximum impulse intensity, are frequently reported as cases of severe muscle damage and corresponding severe rhabdomyolysis [5], in particular after too intense initial WB-EMS application [6].

Depending on the stimulation setting, NMES—as the underlying technology of WB-EMS application—has an impact on deeper muscle groups/layers [2], possibly contributing to the positive effects of WB-EMS on low back pain [7,8,9]. The neuro-muscular principle of (N)EMS is to predominately activate motor neuron axons or intramuscular axonal branches [10], which in turn stimulate the corresponding muscle fibers, allowing innervation of muscles along the neural tracts. Bipolar stimulation techniques, as usually applied during WB-EMS, generate spatially limited current distribution and uniform current density along the current path [11]. Applying magnet resonance imaging (MRI) at the mid-thigh, we [12,13] could show favorable effects on cross sectional volume (CSV) area with a uniform pattern of CSV changes after 16 weeks of WB-EMS in non-athletes. Nevertheless, the standard application of WB-EMS with surface cuff electrodes used for the extremities might have some limitations. First, the optimal positioning of the electrode at the dedicated muscle motor point (motor entry point) might fail considering the multitude of muscles (and muscle motor point) of, for example, the upper leg. Second, impulse intensity is identical for all muscle groups adjacent to the electrode and thus the relative impulse intensity might vary considerably between the stimulated areas, depending on skinfold thickness and current tolerance [14]. In summary, limited information is available as to whether all muscle groups are stimulated in a comparable fashion and maximum depth of stimulation and structural effects. MRI might be a perfect tool to focus on this issue. Morphological MRI has shown its feasibility and its superiority over other imaging modalities to detect exercise-induced muscle changes in several studies [15]. It provides information about changes within the musculature without the need for any invasive procedure, use of contrast media or radiation.

We hypothesize that WB-EMS effects might be more heterogeneous regarding spatial distribution and maximum depth than previously thought.

In order to determine the spatial distribution of a WB-EMS application with respect to volume involvement and stimulation depth using a selected sample of volunteers, we will determine the extent of intramuscular edema at the mid-thigh and lower leg as determined by MRI as a marker of structural effects in the present study.

## 2. Materials and Methods

The present trial should be considered as a pilot study that uses a semi-blinded, randomized controlled design. Briefly, immediately after a baseline MRI scan (to exclude any structural muscle changes) of the mid-femur, we applied one-legged whole-body electromyostimulation to the upper (mid-thigh) and lower leg with the contralateral side serving as a control. A total of 72 h after the intervention, a second MRI scan was acquired to quantify WB-EMS-induced intramuscular edema (Table 1). The study was conducted from May to July 2022 at the Institute of Radiology, University Hospital Erlangen, in close cooperation with the Institute of Medical Physics (IMP), University of Erlangen-Nürnberg (FAU), Germany. The study complied with the Declaration of Helsinki’s “Ethical Principles for Medical Research Involving Human Subjects” and was approved by the ethical committee of the FAU (No. 22-99-B). After detailed information, all participants gave their written informed consent. Table 1 displays the study design.

### 2.1. Participants

Participant recruitment was conducted from May to June 2022. Using personal contacts, eligible persons were contacted by the primary investigator (MG) and informed in detail about the project. After applying the inclusion criteria, (1) 18 years and older, legally capable, (2) no acute or chronic conditions or diseases of the musculoskeletal system, (3) no pharmacologic therapy with impact on the musculoskeletal system, (4) no injuries of the lower limbs, (5) no contraindications for MRI application, (6) no contraindications for WB-EMS application (e.g., pregnancy, diabetes mellitus, epilepsy [16]) and (6) no history of WB-EMS application, six female and five male dentistry students were eligible and willing to participate in the study. However, due to a COVID-19 infection, one participant was unable to conduct the 72 h post MRI scan and was thus not considered for the present analysis.

### 2.2. Randomization and Blinding

The principal investigator conducted randomization of the leg that was stimulated by the WB-EMS application. A coin was tossed to decide whether WB-EMS would be consistently applied to everybody’s right or left leg. Blinding refers to the research assistant that conducted the MRI scan, and the medical imaging expert that analyzed the scan.

### 2.3. Outcomes

Intramuscular edema at the mid-thigh;Intramuscular edema at the lower leg.

### 2.4. Intervention

Participants were asked to maintain their habitual lifestyle but refrain from relevant physical exercise 72 h prior to the intervention and between intervention and follow-up MRI.

### 2.5. WB-EMS Application

Using the present generation of miha-bodytec (Gersthofen, Germany) equipment (Figure 1), WB-EMS was conducted under close medical supervision between 16:00 and 19:00 pm on Sunday afternoon at the Institute of Medical Physics, FAU, Germany. Due to the dedicated MRI protocol and the WB-EMS novice status of the participants, we focused on unilateral WB-EMS of the thigh and lower leg muscles, while the contralateral leg served as a control. In contrast to conventional WB-EMS, we used two electrode cuffs in order to increase the stimulation area (Figure 1b,c) and placed them slightly below the mid-thigh and mid-calf-muscle area and the proximal third of the corresponding muscle belly [17]. Effective electrode area enclosed the entire mid-thigh and lower leg area over a width of (2×) 3.5 cm. Correct placement and appropriate tension of the electrodes were consistently checked and monitored before and during the session. Using a standard WB-EMS protocol [18], 20 min of bipolar current with an impulse frequency of 85 Hz and an impulse width of 350 µs was applied in an intermitted rectangular impulse pattern with 4 s of impulse and 4 s of impulse break. During the impulse phase, voluntary exercises (4–6 sets with 8–12 half squats, lunges, calf raises) for the lower extremities were performed. Impulse intensity was individually adapted for the mid-thigh and lower leg region in close interaction between the participant and the instructor. After 5 min of impulse familiarization, we aimed to generate a rate of perceived exertion at the lower legs and thigh of 7–8 (very hard to hard+ ) on the Borg CR-10 Scale [19]. Importantly, we provided very close supervision, with one instructor responsible for one trainee, and ran a standardized video-guided WB-EMS program to guarantee the close interaction of trainer and trainee, particularly relevant for the adequate regulation of impulse intensity [3].

### 2.6. Semi-Quantitative MRI Analysis

For the semi-quantitative analysis, a radiologist with 8 years’ experience in reading musculoskeletal MRIs evaluated all muscles both at the upper and lower leg. Signal intensity in TIRM images representing intramuscular edema was rated using a modified four-point scale from 0 to 3, based on the scale described by Gerhalter et al. [20] and by Wang et al. [21]: 0—normal appearance; 1—areas of increased signal intensity, 1%–33% of the entire muscle volume; 2—areas of increased signal intensity, 34%–66% of the volume; 3—areas of increased signal intensity, 67%–100% of the volume. All imaging slices of each muscle were considered for the grading.

### 2.7. Statistical Analysis

Due to the general aim and the pilot character of the present study, we applied only descriptive statistical procedures. We used mean values, standard deviation, range and percentage to describe our results.

## 3. Results

### 3.1. Compliance with the WB-EMS Protocol

All participants finished the WB-EMS program described above according to the protocol. However, despite frequent inquiries and encouragement from the trainers to exercise with high intensity (RPE 7–8 at Borg CR 10) two male participants reported perceived exertion rates lower than 7 (5 and 6 at Borg CR 10) and were thus excluded from the analysis. Table 2 presented baseline date of the eight remaining participants with adequate compliance to the protocol.

### 3.2. Baseline MRI Assessment

MRI assessment at baseline did not indicate any muscle edema, injuries or anomalies of the muscle structure in the mid-thigh or mid-lower legs of the participants.

### 3.3. Follow-Up MRI Assessment after 72 h

#### 3.3.1. Upper Leg

Figure 2 shows the transversal T2-weighted MRIs of the upper leg/thigh of three exemplary participants (a–c) of the treated leg (left side) and the contralateral control (without WB-EMS-application) (right side of panel). On the non-stimulated control side, all muscles show a normal appearance without any sign of muscle edema/damage. However, an increase in the signal can be determined at a large variety of mid-thigh (Figure 2a: left side) and lower leg muscles (Figure 3a: left side), reflecting intramuscular edema after 72 h and hence indicating WB-EMS-induced muscle damage.

Descriptive analysis of the data displayed heterogeneous results for the upper leg ROI (Table 3). At the mid-thigh ROI edemas were predominately present at the hamstrings (i.e., m. biceps femoris, semimembranosus and semitendinosus); in two cases edema was observed at the adductor magnus. With respect to the quadriceps, in half of the participants, muscle edema was observed in the m. vastus lateralis, while edema at the vastus medialis, intermedialis and rectus femoris was rare or absent. The depth of the edema (Table 3) ranged from superficial for the m. vastus lateralis in Figure 2 center MRI scan to maximum depth for the m. vastus intermedialis in Figure 2 upper MRI scan. Edema volume at the mid-thigh according to the scale of Gerhalter et al. [20] was consistently ≤33% of the entire muscle.

In summary, we determined grade one muscle damage in superficial but also deeper-lying muscle layers/groups, not only but particularly for the hamstrings (i.e., m. biceps femoris, semimembranosus, semitendinosus); however, we failed to observe a consistently uniform pattern of muscle damage among the eight participants.

#### 3.3.2. Lower Leg

Figure 3 shows the transversal T2-weighted MRIs of the lower leg ROI of three exemplary participants (a–c) of the treated leg (left side) and the contralateral control (without WB-EMS application: right side). WB-EMS-induced muscle damage was slightly more pronounced at the lower leg (Table 4) compared to the mid-thigh area. Edemas at the m. gastrocnemius medialis were verified in all participants. In parallel, all but one or two participants developed muscle edema at the m. gastrocnemius lateralis or the m. soleus, respectively. At the anterior and lateral part of the lower leg, edemas were present at the m. fibularis longus, the extensor digitorum and the tibialis posterior (Table 4). Edema volume varied from one to two thirds of the entire muscle volume (Table 4, Figure 3). Depth of the edema ranged between superficial and maximum depth. Again, we failed to identify a consistently uniform pattern of WB-EMS-induced edemas between the participants.

Importantly, we did not observe differences in number, localization and size of the edemas based on differences in gender, training status or WB-EMS application on the dominant leg or not.

## 4. Discussion

Summarizing our approach, we aimed to determine the spatial distribution of WB-EMS-induced muscle edema at the mid-thigh and lower leg/calf areas as assessed by MRI. First of all, our results do not indicate that WB-EMS failed to simultaneously stimulate and recruit all muscle groups under the cuff electrode surrounding the mid-thigh and lower leg. Moreover, and in line with our expectations, the grade of edema differs between the muscle groups, which suggests that the intensity of (N)EMS-induced muscular contraction was not identical for all muscles. Secondly, superficial (i.e., close to the surface of the skin) and deeper (i.e., closer to bone or internal organs) muscle groups of the mid-thigh or lower leg area were approximately affected in similar fashion (Figure 2 and Figure 3; Table 3 and Table 4), indicating that stimulation of the muscles via surface electrodes also has an effect in deeper parts of the stimulated anatomy. Although rare in human research, the finding that deeper muscle layers/groups were impacted by NMES is not new [23,24,25]. In this context and close to our study, Ogino et al. [25] reported stimulation of deeper m. quadriceps layers after surface NEMS as determined by T2-weighted MRI. This, and particularly the result of the present study, question older research [26] which found that surface-stimulating electrodes typically reach superficial motor units 10–12 mm in close proximity to the electrode’s face. This conclusion, however, might have been confounded by inadequate (low) impulse width [27] or intensity [28] of the NEMS application. In summary, the present study thus contributes to strengthening the evidence that NEMS, applied with surface electrodes, triggers effects on deeper muscle layers of the mid-thigh and lower legs/calf area.

Apart from the finding that WB-(N)EMS application enables intense stimulation of deeper muscle groups of the thigh and particularly the lower leg area, the spatial distribution of muscle edema as a marker of intense stimulation is important. Applying the identical protocol to all participants, we detected quite a heterogeneous pattern of increased MRI-signal-indicating areas of high muscular activation among the participants. Differences related to gender, skinfold/subcutaneous thickness [29], leg shape, which influence the contact of the electrodes to the tissue and the current flow, individual location of muscle motor points [17] and specificity of fiber composition [30] might have contributed to this finding. Furthermore, different stimulus thresholds of the muscles with resulting differences in contraction strength, as well as differences in the stress resistance of the muscles with resulting variations in the level of damage of muscle structures may have been a reason for the less homogeneous (non-selective) pattern in our cohort of novice WB-EMS applicants [23,31]. However, we are unable to conclude whether the contraction strength of the ischiosural muscles was higher compared to quadriceps muscles due to a lower stimulus threshold, or whether a lower mechanical stress resistance of this muscle group leads to more severe muscular damage when exposed to the same loading. With respect to the mid-thigh area, WB-EMS-induced edema was not only, but predominately, detected at the hamstring group, a muscle compartment that particularly suffers from a sitting lifestyle [32]. Accordingly, one explanation would be that due to the corresponding degeneration, the hamstring muscle group reacts particularly sensitively to electrical stimuli, while the m. quadriceps is exposed to greater mechanical stress in everyday life (especially when walking down stairs) and could therefore be mechanically more resilient compared to the hamstrings.

Inspecting lower leg muscle groups, the number and volume of the edemas were even larger compared with the mid-thigh area. Comparable to the hamstrings muscle groups, a chair-based sitting position generates very low muscular activation in the lower leg muscles (as assessed by EMG) [33]. In everyday life, the calf muscles, unlike the m. quadriceps, are used almost exclusively concentrically, with the muscle connective tissue being subjected to lower levels of stress. Correspondingly, the calf area might be particularly sensitive to dedicated muscular strain applied by NEMS. However, we have no clear explanation for the result that the anterior lower leg muscles (in contrast to the calf muscles) did not exhibit pronounced edema. We speculate that differences in the fit of the electrode, variations in the stimulation threshold or differences in muscle strain in everyday life (when walking, the foot lifters work eccentrically during the initial stance phase), might have contributed to this result.

Some particularities and limitations of the present research must be taken into account when interpreting the clinical significance of our findings. (1) In the present study, we focused on edema as a structural marker of NEMS effects. Future research applying MRI may include more advanced imaging techniques like diffusion tensor imaging and other more sophisticated techniques like sodium MRI [15]. (2) We refrain from a sophisticated statistical procedure due to the pilot character of the study with its predominately descriptive character. Pre-intervention MRI assessments and the approach to use the right leg as a non-(N)EMS control was applied to ensure that edemas can be exclusively attributed to the intense WB-EMS application. (3) We recruited a cohort of dentistry students without athletic background and any former local or WB-(N)EMS application. This approach was selected in order to generate a pronounced and visible WB-EMS-induced effect on muscle groups of the mid-thigh and lower leg. Nevertheless, exercise habits vary considerably between our participants (Table 2), thus one may argue that less physically trained participants suffer from a higher degree of edema. We addressed this issue; however, we did not observe corresponding differences in the number or volume of edemas. (4) Being aware that applying high impulse intensity during WB-EMS first application might lead to pronounced negative side effects [6,34], the session was conducted under close medical supervision. (5) Considering the pronounced repeated bout effect observed for WB-EMS-induced rhabdomyolysis [35,36] the question arises whether the present outcomes would be similar when experienced WB-EMS applicants are included. (6) Following WB-EMS standard protocols [1], low–moderate intensity voluntary exercises were performed during the impulse phase of the WB-EMS (4 s, with 4 s of rest). One may argue that the effects of voluntary exercise may confound our results that focus on WB-EMS. However, reviewing the contralateral side not addressed by WB-EMS (but voluntary exercises), we observed no signs of edemas/increased MRI signals. (7) Finally, we did not apply a “purebred WB-EMS approach” while limiting stimulation on four electrodes at the lower extremities [3]. However, considering the large amount of upper and lower muscles under the electrodes, we can hardly summarize our approach under (local) EMS.

## 5. Conclusions

In summary, the present study indicates that WB-EMS enables intense stimulation of both superficial and deep muscle groups of the thigh and lower leg. However, due to the uniform stimulation by the cuff electrodes the spatial distribution of muscular effects is heterogeneous. Future research is needed to determine the muscular effects of WB-EMS and the potential role of imaging to address this issue.

## Figures and Tables

**Figure 1 sensors-22-10017-f001:**
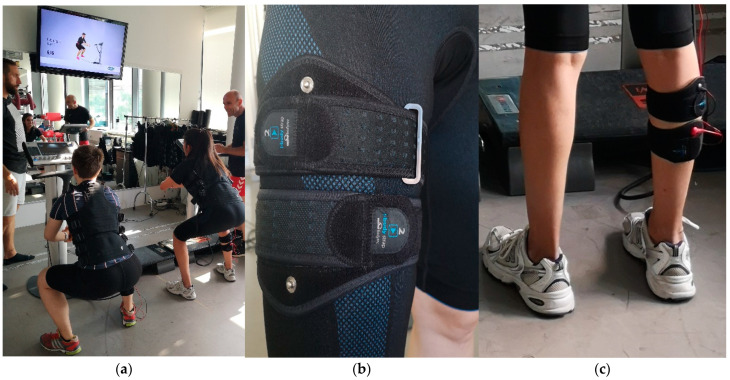
(**a**) Supervised video-guided WB-EMS application with two trainees supervised by two instructors (left image) (**b**) electrode placement at the mid-thigh (center image) (**c**) electrode placement at the mid-calf (right image).

**Figure 2 sensors-22-10017-f002:**
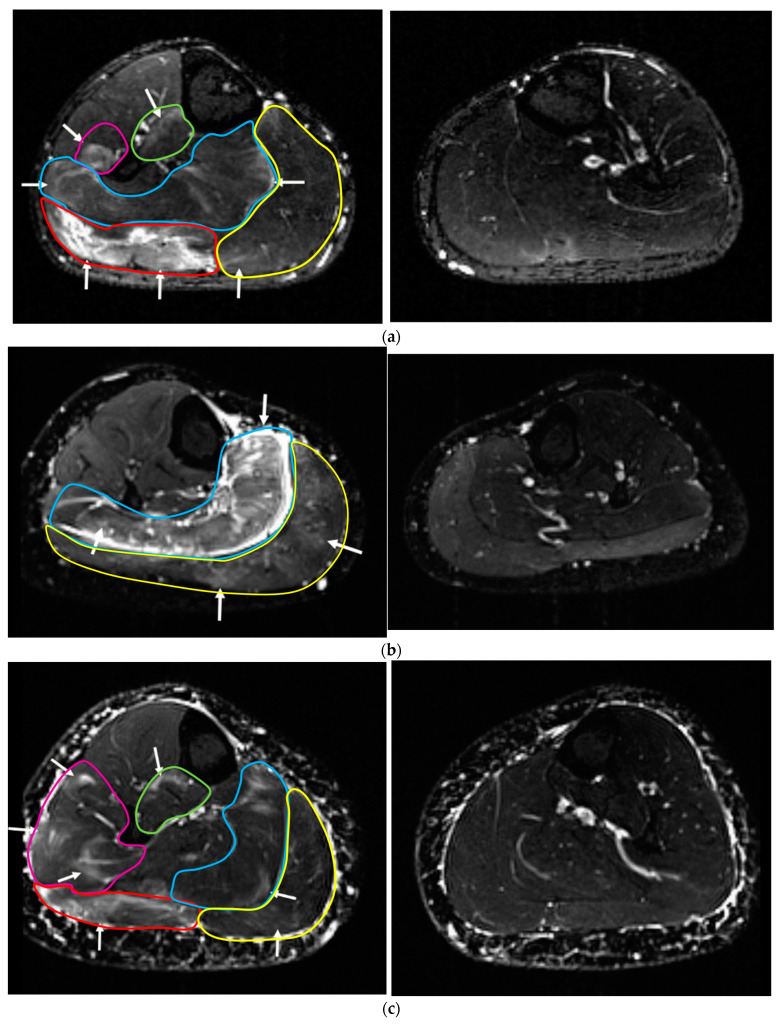
Muscle status with (left side) or without (right side) WB-EMS application at the mid-thigh region of interest (ROI) of three participants (**a**–**c**) as determined by magnetic resonance imaging. (**a**) Intramuscular edema at m. vastus lateralis, vastus intermedius, vastus medialis, long head and short head of biceps femoris, semitendinosus, semimembranosus (treated leg). All muscles show a normal appearance without any sign of WB-EMS induced muscle edema (control leg). (**b**) Intramuscular edema at m. vastus lateralis, long head of biceps femoris, and semimembranosus (treated leg). All muscles show a normal appearance without any sign of WB-EMS-induced muscle edema (control leg). (**c**) Intramuscular edema at m. biceps femoris caput longum, semitendinosus, semimembranosus, adductor magnus (treated leg). All muscles show a normal appearance without any sign of WB-EMS-induced muscle edema (control leg).

**Figure 3 sensors-22-10017-f003:**
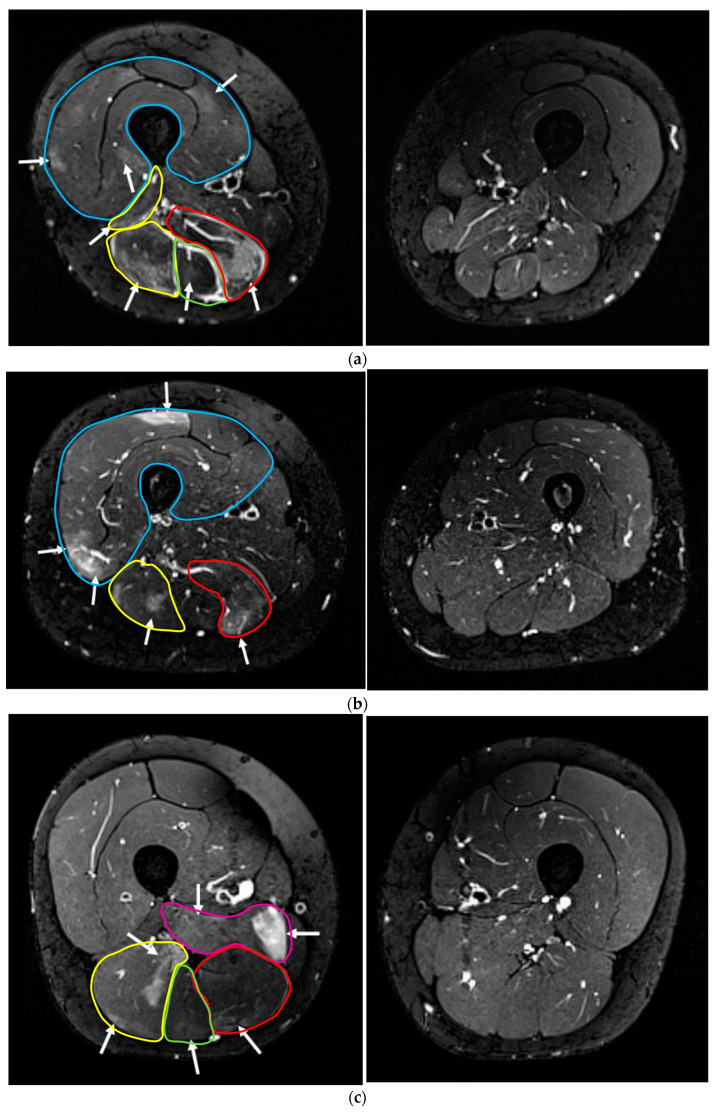
Muscle status with (left side) or without (right side) WB-EMS application at the lower leg ROI of three participants (**a**–**c**) as determined by magnetic resonance imaging. (**a**) Intramuscular edema at m. gastrocnemius medialis und lateralis, soleus, fibularis longus, tibialis posterior (treated leg). All muscles show a normal appearance without any sign of WB-EMS-induced muscle edema (control leg). (**b**) Intramuscular edema at m. gastrocnemius medialis and soleus (treated leg). All muscles show a normal appearance without any sign of WB-EMS-induced muscle edema (control leg). (**c**) Intramuscular edema at m. gastrocnemius medialis und lateralis, soleus, fibularis longus and tibialis posterior (treated leg). All muscles show a normal appearance without any sign of WB-EMS-induced muscle edema (control leg).

**Table 1 sensors-22-10017-t001:** Study design of the present project.

MRI-Scan I	WB-EMS Application	Rest Period	MRI-Scan II
Baseline	Immediately after MRT	After WB-EMS—72 h	72 h post

**Table 2 sensors-22-10017-t002:** Baseline characteristics of the participants.

Variable	Women		Men	
	MV ± SD, Number	Range	MV ± SD, Number	Range
Gender	5	---------	3	----------
Age [years]	26.4 ± 2.8	24–31	24.0 ± 1.0	23–25
Body height [cm] ^1^	166.5 ± 3.5	163–171	178.7 ± 12.2	168–192
Body mass [kg] ^2^	59.2 ± 7.0	50.7–69.9	77.0 ± 12.9	67.2–91.6
Total body-fat [%] ^1^	21.5 ± 2.6	19.4–25.8	20.2 ± 5.5	13.8–26.8
Physical activity ^3^	4.4 ± 0.5	4–5	3.3 ± 1.5	2–5
Physical fitness ^3^	4.4 ± 0.6	4–5	4.7 ± 0.6	4–5
Exercise [sessions/w] ^4^	2.1 ± 1.9	0–5	2.2 ± 1.3	1–4
Diseases [*n*] ^4^	0	------------	0	------------
Medication [*n*] ^4^	0	------------	0	------------
Smokers [*n*] ^4^	0	------------	0	------------

^1^ As determined by a Holtain stadiometer (Crymych Dyfed., UK); ^2^ as determined by Bio Impedance Analysis (Inbody 770, Seoul, Republic of Korea); ^3^ based on a questionnaire with a scale from (1) “very low” to (7) “very high” [22]; ^4^ as determined by questionnaire and verified by personal interview.

**Table 3 sensors-22-10017-t003:** Descriptive characteristics of (N)EMS-induced edemas at the mid-thigh.

Compartment of the Edema	Participants with Edema (Total: *n* = 8)	Maximum Depth of the Edema from Skin (mm) MV ± SD, (Range)	Intramuscular Edema Volume (Grading) MV (Range)
m. vastus lateralis	4	29 ± 9; 19–39	1 (1)
m. vastus medialis	2	25; 47	1 (1)
m. vastus intermedialis	1	61	1
m. rectus femoris	0	-------	--------
m. biceps femoris	6	50 ± 12; 39–72	1 (1)
m. semimembranosus	5	31 ± 6; 23–37	1 (1)
m. semitendinosus	3	37 ± 6; 32–44	1 (1)
m. adductor magnus	2	46; 68	1 (1)

**Table 4 sensors-22-10017-t004:** Descriptive characteristics of (N)EMS-induced edemas at the lower leg.

Compartment of the Edema	Participants with Edema (Total: *n* = 8)	Maximum Depth of the Edema from Skin (mm) MV ± SD, (Range)	Intramuscular Edema Volume Grading MV (Range)
m. gastrocnemius medialis	8	24 ± 5; (18–33)	1 (1)
m. gastrocnemius lateralis	7	26 ± 4; (21–31)	1.3 (1–2)
m. soleus	6	35 ± 14; (19–51)	1.2 (1–2)
m. peroneus longus	4	31 ± 4; (25–35)	1 (1)
m. extensor digitorum	1	25	1 (1)
m. tibialis posterior	2	42; 56	1 (1)

## Data Availability

The data that support the findings of this study are available from the corresponding author (WK) upon reasonable request.

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
