# Peer review of "Spatial Distribution of Muscular Effects of Acute Whole-Body Electromyostimulation at the Mid-Thigh and Lower Leg—A Pilot Study Applying Magnetic Resonance Imaging"

_sensors, 2022, doi:10.3390/s222410017_

Round 1
Reviewer 1 Report
See attached file

Reviewer 2 Report
This paper describes spatial distribution of electromyostimulation (WB-EMS)-induced muscle intramuscular edema using magnetic resonance imaging (MRI). The manuscript is well-written, the muscle status characterization is thorough, and the figures are appealing. Consequently, I believe that the manuscript is appropriate for sensors.
Comments: The authors need to explain the advantages of their magnetic resonance imaging (MRI) methods over other methods in the literature. A table or radar chart was recommended to be added in the manuscript.
The amended paper can be considered for publication.
Round 2
Reviewer 1 Report
The authors have correctly addressed all my comments. Thank you very much for the efforts and congratulations for the work.